# Transcriptome and Metabolome Analysis of Color Changes during Fruit Development of Pepper (*Capsicum baccatum*)

**DOI:** 10.3390/ijms232012524

**Published:** 2022-10-19

**Authors:** Yu Zhang, Huangying Shu, Muhammad Ali Mumtaz, Yuanyuan Hao, Lin Li, Yongjie He, Weiheng Jin, Caichao Li, Yan Zhou, Xu Lu, Huizhen Fu, Zhiwei Wang

**Affiliations:** 1Key Laboratory for Quality Regulation of Tropical Horticultural Crops of Hainan Province, School of Horticulture, Hainan University, Haikou 570228, China; 2Sanya Nanfan Research Institute, Hainan University, Sanya 572025, China; 3Hainan Yazhou Bay Seed Laboratory, Sanya 572025, China

**Keywords:** pepper, flavonoids, carotenoids, transcriptome, metabolome

## Abstract

Fruit color is one of the most critical characteristics of pepper. In this study, pepper (*Capsicum baccatum* L.) fruits with four trans-coloring periods were used as experimental materials to explore the color conversion mechanism of pepper fruit. By transcriptome and metabolome analysis, we identified a total of 307 flavonoid metabolites, 68 carotenoid metabolites, 29 DEGs associated with flavonoid biosynthesis, and 30 DEGs related to carotenoid biosynthesis. Through WGCNA (weighted gene co-expression network analysis) analysis, positively correlated modules with flavonoids and carotenoids were identified, and hub genes associated with flavonoid and carotenoid synthesis and transport were anticipated. We identified Pinobanksin, Naringenin Chalcone, and Naringenin as key metabolites in the flavonoid biosynthetic pathway catalyzed by the key genes *chalcone synthase* (*CHS CQW23_29123*, *CQW23_29380*, *CQW23_12748*), *cinnamic acid 4-hydroxylase* (*C4H CQW23_16085*, *CQW23_16084*), *cytochrome P450* (*CYP450 CQW23_19845*, *CQW23_24900*). In addition, *phytoene synthase* (*PSY CQW23_09483*), *phytoene dehydrogenase* (*PDS CQW23_11317*), *zeta-carotene desaturase* (*ZDS CQW23_19986*), *lycopene beta cyclase* (*LYC CQW23_09027*), *zeaxanthin epoxidase* (*ZEP CQW23_05387*), *9-cis-epoxycarotenoid dioxygenase* (*NCED CQW23_17736*), *capsanthin/capsorubin synthase* (*CCS CQW23_30321*) are key genes in the carotenoid biosynthetic pathway, catalyzing the synthesis of key metabolites such as Phytoene, Lycopene, β-carotene and ε-carotene. We also found that transcription factor families such as p450 and NBARC could play important roles in the biosynthesis of flavonoids and carotenoids in pepper fruits. These results provide new insights into the interaction mechanisms of genes and metabolites involved in the biosynthesis of flavonoids and carotenoids in pepper fruit leading to color changes in pepper fruit.

## 1. Introduction

The fruit color is the most important quality trait of pepper because the pigments that impart color are associated with fruit nutrition, health, and flavor [1]. Therefore, pepper fruit color has long been of great interest to breeders and consumers [2]. *Capsicum* fruits can show a range of colors from white to deep red [3]. It has been reported that the fruit color of chili pepper is determined by the relative content of pigment substances such as chlorophylls, carotenoids, and flavonoids in the fruit of chili pepper [4].

Carotenoids are both specialized and primary metabolites, and their accumulation is complexly regulated throughout the plant life cycle in response to developmental and environmental signals [5]. Carotenoids are synthesized in plants via the mevalonate pathway in the cytoplasm and the Mevalonate pathway (MEP) pathway in the plastid [6]. Its biosynthesis begins with isopentenyl diphosphate (IPP) and dimethylallyl diphosphate (DMAPP). Geranylgeranyl diphosphate synthase (GGPPS) adds three IPP molecules to DMAPP to generate Geranylgeranyl pyrophosphate (GGPP) [5,7]. Flavonoids are structurally diverse, and more than 8000 flavonoids have been isolated and identified from plants, including chalcones, chrysanthones, flavanones, flavones, isoflavones, dihydroflavonols, flavonols, colorless anthocyanins, anthocyanidins and proanthocyanidins [2,8]. In plants, flavonoids are synthesized by the deamination of phenylalanine in the phenylpropane pathway [9]. Phenylalanine is converted to cinnamic acid catalyzed by phenylalanine deaminase (PAL), and cinnamic acid CoA-ester and malonyl-CoA are reduced to naringin and chalcone catalyzed by chalcone synthase (CHS), which is then isomerized to flavanone by chalcone flavanone isomerase (CHI) [10]. Flavanone 2-hydroxylase (*ZmF2H1*) converts flavanones to the corresponding 2-hydroxyflavones [11], Hydroxy flavone intermediates dehydrated into flavonoids [12], Naringenin is catalyzed by different enzymes to form different types of flavonoids [13].

In higher plants, the biosynthesis of carotenoids and flavonoids is not only regulated by related genes but also involves many transcription factors. Many transcription factors (TFs) have been shown to affect carotenoid accumulation by regulating fruit ripening, ethylene biosynthesis, photomorphogenesis, and other processes [14]. In tomato, several transcription factors such as *MYB*, *CNR*, *RIN*, *TAGL1*, and *SGR* are involved in regulating ripening and thus carotenoid accumulation in the fruit [15].

In this study, differentially expressed genes involved in carotenoid and flavonoid metabolic pathways were analyzed by comparative transcriptomics. The hub genes and transcription factors associated with carotenoids and flavonoids were identified using Weighted gene co-expression network analysis (WGCNA). We also used the combined metabolomic and transcriptomic analysis to explore the interactions between related metabolites and genes in the process of pepper color change. The results of this study provide essential data on the metabolic pathways of carotenoids and flavonoids regulating the regulatory mechanism of pepper fruit color change from the initial white to the final red stage.

## 2. Results

### 2.1. Physiological Index Analysis

Carotenoids, flavonoids and chlorophylls, as important pigments in plants, play crucial roles in the formation of pigments in pepper fruits. We determined the carotenoid, flavonoid, and chlorophyll contents in the fruits of HNUCB0081 at four developmental stages (Figure 1). Carotenoid, flavonoid, and chlorophyll contents were significantly different within pepper fruits at four periods. The carotenoid and flavonoid content were the lowest in the white fruit and highest in the yellow fruit (Figure 1). Chlorophyll content were highest in the white fruit and lowest in the red fruit, with a decreasing trend (Figure 1). Interestingly, we found higher content of flavonoids and carotenoids within the pepper fruit and their levels varied during the different growth periods, so we followed up with a focused analysis of flavonoids and carotenoids.

### 2.2. Metabolome Analysis

To further confirm the metabolite changes in pepper fruits during the color change process, the carotenoid and flavonoid metabolomics assay was carried out using UPLC-ESI-MS/MS on 4 periods of pepper HNUCB0081.

We detected 307 flavonoid metabolites in 12 samples, including 118 flavonols, 89 flavones, 40 dihydroflavones, 17 chalcones, 14 flavanols, 10 isoflavones, 10 dihydroflavonols, 6 ellagitannins, 2 proanthocyanidins and 1 other flavonoid (Appendix A). In the thermogram, the metabolites in CbW, CbY, CbO and CbR showed similar accumulation trends (Appendix A). The metabolites luteolin-7-O-(6″-malonyl) glucoside, 4,4′-dihydroxy-2,6-dimethoxydihydrochalcone, and Castanoside A [Kaempferol-3-O-(6″-*p*-coumaroyl) mannoside], etc. accumulated significantly during the white fruit period; the metabolites Isorhamnetin, 3′-Methoxy-3,4′,5,7-Tetrahydroxyflavone, Quercetin-3-O-rutinoside-7-O-rhamnoside, and 5,7,3′,4′,5′-Pentahydroxydihydro flavone accumulated significantly during the yellow fruit period; the metabolites Naringenin (5,7,4′-Trihydroxyflavanone), Naringenin chalcone, and Butin, 7,3′,4′-Trihydroxyflavanone accumulated significantly during the orange fruit period; the metabolites Quercetin-3-O-[2″-O-(6″′-*p*-coumaroyl) glucosy1] rhamnoside, Quercetin-3-O-rutinoside-7-O-glucoside, and Sachaloside IV accumulated significantly during the red fruit period.

In the 3 adjacent comparative combinations of CbW-vs-CbY, CbY-vs-CbO, and CbO-vs-CbR, there were 148, 84 and 122 differential metabolites with a unique number of differential metabolites of 61, 11 and 48, respectively, 26 differential metabolites were present in all three comparative combinations (Figure 2A). Among them, 61 and 87 differential metabolites were upregulated and downregulated in CbW-vs-CbY, 77 and 7 differential metabolites were upregulated and downregulated in CbY-vs-CbO, 16 and 106 differential metabolites were upregulated and downregulated in CbO-vs-CbR, respectively (Figure 2C).

We detected a total of 68 carotenoid metabolites in 12 samples, including 7 carotenoids (β-carotene, hexahydro lycopene, lycopene, α-carotene, γ-carotene, ε-carotene, and octahydro lycopene) and 61 lutein (carotenoids, violet xanthin, capsaicin, and among others) (Appendix A). We used hierarchical cluster analysis to classify the accumulation and variation models of carotenoid metabolites in fruit samples from four periods of pepper (Appendix A). In the thermogram, the metabolites in CbW, CbY, CbO and CbR showed similar accumulation trends. Among them, lutein palmitate, lutein, lutein dimyristate, and neoxanthin accumulated significantly in the white fruit period; violaxanthin myristate accumulated significantly in the yellow fruit period; Metabolites such as rubixanthin palmitate, β-cryptoxanthin palmitate and zeaxanthin were significantly expressed in the orange fruit period. Metabolites such as anthaxanthin, ε-carotene and β-citraurin were significantly expressed in the red fruit period. β-cryptoxanthin laurate, violaxanthin-myristate-laurate and violaxanthin-myristate-caprate were more significantly expressed in orange and red fruits. The carotenoid content was 32, 35 and 16 in the three adjacent combinations of CbW-vs-CbY, CbY-vs-CbO, and CbO-vs-CbR, respectively, and seven of the differential metabolites (5,6 epoxylated luteolin-haemate-palmitate, β-citrulline, and zeaxanthin, among others) were present in all three combinations. There were five, five, and two differential metabolites specific to each of CbW-vs-CbY, CbY-vs-CbO and CbO-vs-CbR, respectively (Figure 2B). Among them, there were 26 and 6 up- and down-regulated differential metabolites in CbW-vs-CbY, respectively; 33 and 2 up- and down-regulated differential metabolites in CbY-vs-CbO, respectively; 7 up- and 9 down-regulated differential metabolites in CbO-vs-CbR. (Figure 2D).

### 2.3. Differential Metabolite Enrichment Analysis

Three comparative combinations of flavonoid differential metabolites of CbW-vs-CbY, CbY-vs-CbO, and CbO-vs-CbR were co-enriched to five pathways (ko00944, ko00944, ko00943, ko01100 and ko01110) (Figure 3A–C). Among the first 20 metabolites of the CbW-vs-CbY combination, 6 major metabolite classes were included: flavonols (10), chalcones (2), flavanonols (2), flavanols (1), fsoflavones (1) and flavonoids (4). The top 20 metabolites in the CbY-vs-CbO combination contained 4 major metabolite classes, with the most flavonols (9) and flavones metabolites (9), followed by flavanones (1) and flavanols (1). The top 20 metabolites of the CbO-vs-CbR combination contained 6 major metabolite classes, of which 11 metabolites belonged to flavonols and 4 metabolites to flavones (Appendix A).

Carotenoid differential metabolites in the three comparison groups, CbW-vs-CbY, CbY-vs-CbO, and CbO-vs-CbR, were enriched to four pathways, ko00906, ko01100, ko01110 and ko01240, with the highest number of metabolites enriched in the ko00906 pathway (Figure 3D,F). Notably, seven metabolites were present in all three comparative combinations, including β-carotene, 5, 6-epoxylated luteolin-palmitate and β-citrulline (Appendix A).

### 2.4. Transcriptome Results Analysis

To investigate the molecular mechanisms involved in controlling the transformation of metabolites within pepper fruits, thus leading to the color change of pepper fruits, we sequenced the transcriptome of fruits from four periods of pepper HNUCB0081.

After removing the adapter and low-quality sequences, each library received 39,171,874–42,718,356 clean reads (Appendix A). These clean reads were mapped to the reference genome with matches ranging from 70.96–96.34% and overall sequencing error rates of 2–3% for both data, with Q20 reads ranging from 97.98–98.68% and Q30 reads ranging from 93.74–95.67%, all with GC content above 40% (Appendix A).

A total of 33,056 DEGs were identified by differential expression analysis between 14 periods of the fruit of pepper HNUCB008, with the number of differentially expressed genes (DEGs) ranging from 1727 to 9272 between sample groups (Appendix A). The principal component analysis (PCA) results showed that fruit samples from different periods could be clearly distinguished on PCA, and different metabolites could be distinguished on PC2 composition. The heat map showed a relatively high correlation between biological replicates of the same group of samples, indicating better replication within the selected sample groups (Appendix A).

In 3 comparative combinations of CbW-vs-CbY, CbY-vs-CbO and CbO-vs-CbR at three adjacent developmental periods, 5285 (3366 up-regulated and 1919 down-regulated), 5393 (2809 up-regulated and 2584 up-regulated) and 1727 (1147 up-regulated and 580 down-regulated) DEGs were identified, respectively. Among all up- and down-regulated genes, 28 and 12 genes were expressed in all four periods, 199 and 92 genes were expressed in both CbW-vs-CbY and CbO-vs-CbR, 37 and 13 genes were expressed in both CbY-vs-CbO and CbO-vs-CbR, and 607 and 392 genes were expressed in both CbW-vs-CbY and CbY-vs-CbO, respectively (Appendix A). The results show that transcriptome changes were induced in the pepper pericarp at different developmental periods of the pepper.

### 2.5. Functional Analysis of Differential Genes

To verify the biological functions of the differential genes in the peel of peppers from four periods, KEGG and GO functional enrichment analyses were performed on the differential genes. A total of 946 genes were annotated on 56 clades of biological process (BP), cellular component (CC) and molecular function (MF) in the three comparison combinations of CbW-vs-CbY, CbY-vs-CbO and CbO-vs-CbR. Where carbohydrate metabolic process (GO: 0005975) in BP, Extrinsic component of membrane (GO: 0019898), thylakoid (GO: 0042651), photosystem (GO: 0009521), photosynthetic membrane (GO:0034357), photosystem II oxygen-evolving complex (GO:0009654), thylakoid part (GO:0044436), thylakoid membrane (GO:0042651) and oxidoreductase complex (GO:1990204) in CC, As well as the transferase activity in MF, transferring acyl groups (GO: 0016746) was significantly enriched in all three combinations (Appendix A). Among all the up-regulated DEGs, 197, 24, 52, 48, 50, 26, 52, 28, 26 and 81 DEGs annotated the above 10 GO terms, respectively, among all the down-regulated DEGs, 133, 2, 8, 8, 8, 3, 8, 3, 3 and 55 DEGs annotated the above 10 GO terms, respectively. The GO enrichment results indicate that the metabolic processes, catalytic activity and cellular processes contained in the GO terms play important roles in regulating the formation and transformation of pepper fruit color.

KEGG annotation results showed that Flavonoid biosynthesis (sly00941) and Carotenoid biosynthesis (sly00906) were significantly enriched in all three combinations, CbW-vs-CbY, CbY-vs-CbO and CbO-vs-CbR. (Appendix A). A total of 29 DEGs were identified in the flavonoid biosynthetic pathway, including key genes such as *CHS* (*chalcone synthase*; *CQW23_29123*, *CQW23_29380* and *CQW23_12748*), *C4H* (*cinnamic acid 4-hydroxylase*; *CQW23_16084* and *CQW23_16085*); 30 DEGs were identified in the carotenoid biosynthetic pathway, including key genes including *PSY* (*phytoene synthase*; *CQW23_09483*), *LYC* (*lycopene beta cyclase*; *CQW23_09027*) and *CCS* (*capsanthin/capsorubin synthase*; *CQW23_30321*). Thus, fruit color formation and transformation in peppers is a biological process involving the combined action of multiple pathways.

### 2.6. Expression of Differential Genes in Carotenoid and Flavonoid Biosynthetic Pathways

In the carotenoid biosynthetic pathway, 13 genes including *CQW23_04804*, *CQW23_06623*, and *CQW23_10307* were up-regulated and 7 genes including *CQW23_11317*, *CQW23_29327* and *CQW23_19986* were down-regulated in the combination of CbW-vs-CbY. In CbY-vs-CbO, 7 genes including *CQW23_24636*, *CQW23_06623,* and *CQW23_10308* were up-regulated, and 9 genes including *CQW23_07725*, *CQW23_05387* and *CQW23_30321* were down-regulated. 7 genes, including *CQW23_09483*, *CQW23_14283* and *CQW23_29956*, were up-regulated in CbO-vs-CbR, and only 1 gene, *CQW23_01678*, was down-regulated. Furthermore, 11 genes including *CQW23_09483*, *CQW23_14283* and *CQW23_10307* were expressed in two combinations at the same time, and *CQW23_07725* was expressed in all three combinations (Appendix A).

In the biosynthesis pathway of flavonoids, 13 genes including *CQW23_24900*, *CQW23_29380* and *CQW23_19845* were up-regulated in the CbW-vs-CbY combination, and 3 genes including *CQW23_09726*, *CQW23_10085* and *novel.807* were down-regulated. In CbY-vs-CbO combination, *CQW23_29123*, *CQW23_04685* and *novel.811* were up-regulated, and 9 genes including *CQW23_29379*, *CQW23_32878* and *CQW23_19845* were down-regulated. *CQW23_24900*, *CQW23_19845* and *CQW23_32878*, were expressed in both CbW-vs-CbY and CbY-vs-CbO (Appendix A). The pathway was not enriched in the CbO-vs-CbR combination.

It was further demonstrated that the carotenoid and flavonoid biosynthetic pathways play an important role in the transformation process of peel formation in peppers, and that these two pathways are regulated by several genes, which change during different color change periods.

### 2.7. Transcription Factor Analysis

The differentially expressed genes in the three comparative combinations of CbW-vs-CbY, CbY-vs-CbO, and CbO-vs-CbR in this study were annotated to a total of 4695 transcription factors belonging to 457 TF families, which mainly included Pkinase (345), NBARC (191), p450 (182) and UDPGT (78). Among the 17 major TF families with high numbers, the number of up-regulated and down-regulated genes in the 3 comparative combinations was 1041 and 574, respectively, and there was a decreasing trend in the number of genes in the 3 combinations (Figure 4A). In CbW-vs-CbY, the number of up-regulated genes was significantly higher than the number of down-regulated genes in the 15 families except AAA and PP2C (Figure 4B). In CbY-vs-CbO, there are little difference in the number of up and down-regulated genes (Figure 4B). In CbO-vs-CbR, the number of up-regulated genes was significantly higher than the number of down-regulated genes, and the number of up and down-regulated genes in this combination was significantly lower than that in both CbW-vs-CbY and CbY-vs-CbO combinations (Figure 4B). The Pkinase family had the highest number of genes among the three compared combinations, in addition to the p450, NBARC, Pkinase_Tyr and Myb_DNAbinding families (Figure 4A,B), which may play important roles in pepper fruit color change.

### 2.8. Gene Co-Expression Network Analysis

#### 2.8.1. Module Selection

To investigate the gene regulatory network of related pigment synthesis in pepper fruit, we performed a weighted gene co-expression network analysis (WGCNA) using non-redundant DEGs. These non-redundant DEGs are clustered into 24 main branches, each representing a module (marked with a different color) (Figure 5A), where modules are clusters of genes with high relatedness and genes within the same module are co-expressed.

Subsequently, we analyzed three horticultural physiological indicators of module-trait correlation at four developmental stages, CbW, CbY, CbO and CbR (Figure 5B). The results showed that the pink module was positively correlated with carotenoid and flavonoid content in pepper fruit, and the yellow module was positively correlated with chlorophyll content in pepper fruit. To determine the expression patterns of the genes in the pink and yellow modules, we used the FPKM values of the genes in the modules to create heat maps for analysis. Heat map results showed that genes co-expressed in the pink module were expressed in the yellow and red fruit periods, and genes co-expressed in the yellow module were expressed in the white and yellow fruits (Figure 5C,D).

#### 2.8.2. Screening for Transcription Factors That Regulate Pigment Synthesis by WGCNA

Based on the connectivity of gene nodes, genes can be further divided into hub genes (with extremely high connectivity) and non-hub genes [16]. We specified the genes in the top 20 genes of connectivity as hub genes. Based on the degree of connectivity within the module, we screened the top 20 genes to be considered as central genes. Among the first 20 central genes of the pink module, we identified 7 transcription factors, namely *NBARC* (*CQW23_18627*), *Pribosyltran* (*CQW23_23290*), *CPSase_sm_chain* (*CQW23_11908*), *Pkinase_Tyr* (*CQW23_ 18567*), *PfkB* (*CQW23_27627*), *NAD_binding_1* (*CQW23_04069*), and *ILVD_EDD* (*CQW23_28591*). Among the first 20 central genes in the yellow module, we identified 9 transcription factors, *Glyco_hydro_17* (*CQW23_20937*), *Glycos_transf_1* (*CQW23_00121*), *zf-CCCH* (*CQW23_15524*), *Pro_isomerase* (*CQW23_20849*), *UDPGT* (*nov.753*), *ADH_zinc_N* (*CQW23_29424*), *MBD* (*CQW23_26228*), *GST_N* (*CQW23_17725*) and *Pkinase* (*CQW23_26424*). These transcription factors act as highly linked central genes and may have a regulatory role in the formation of pepper fruit color.

To further understand the transcription factors involved in the regulation of pepper pigments, correlation network analysis was constructed using the central genes in the pink and yellow modules and the top 10 genes corresponding to the weight value of each gene in the network node relationships (200 genes in total), respectively. The results showed that, in the pink module, 103 genes were highly correlated with carotenoid synthesis (edge weight ≥ 0.155), including hub genes (Figure 6A, Appendix A); in the yellow module, 104 genes were highly correlated with chlorophyll synthesis (edge weight ≥ 0.308), and in addition to hub genes, *ubiquitin* (*CQW23_13952*), *HLH* (*CQW23_11941*) and *SBP* (*CQW23_26319*), which are non-Hub transcription factor genes, were also associated with the regulation of chlorophyll synthesis (Figure 6B, Appendix A).

### 2.9. Association Analysis of Transcriptome and Metabolome

#### 2.9.1. Correlation Analysis of Transcriptome and Metabolome

To determine the correlation between differential metabolites and differential genes within four periods of pepper pericarp, we performed differential genes Pearson correlation analysis for differential genes and differential metabolites, (Pearson correlation coefficient >0.8 or <−0.8, *p*-value < 0.05). The results showed that 10 genes, including *novel.1236*, *CQW23_25223* and *CQW23_24900*, were up-regulated in CbW-vs-CbY. 11 metabolites, including Naringenin Chalcone (pme2960), Hesperetin (MWSHY0049) and Luteolin (pme0088), were down-regulated. The gene Novel.1236 was negatively correlated with mws0914, mws2118 and mws1422, and positively correlated with Luteolin. Gene *CQW23_04982* was positively correlated with the metabolite Hesperetin. The gene *CQW23_24900* was positively correlated with metabolites Homoeriodictyol (mws1033), Phlorizin Chalcone (Lmlp006175), Luteolin (pme0088) and Hesperetin (MWSHY0049). Gene *CQW23_16084* was positively correlated with metabolite Luteolin (pme0088) (Figure 7A). In CbY-vs-CbO, gene *CQW23_24900* was positively correlated with the metabolite Homoeriodictyol, Phlorizin Chalcone and Luteolin. Gene CQW23_16084 was positively correlated with metabolite Luteolin (Figure 7B). In CbO-vs-CbR, gene *CQW23_24900* was positively correlated with the metabolites Homoeriodictyol, Luteolin, Phlorizin Chalcone and Hesperetin. Genes such as *CQW23_24900* and *CQW23_16084* and metabolites such as Luteolin and Phlorizin Chalcone are all involved in regulation in the three combinations (Figure 7C). These results indicated that *CQW23_24900* and *CQW23_16084* genes and metabolites such as Luteolin and Phlorizin Chalcone were involved in the whole color transformation process of pepper.

#### 2.9.2. Transcriptome and Metabolome Pathway Analysis

We further plotted the enrichment histograms by performing KEGG enrichment analysis on the transcriptome and metabolome. The results showed that only one pathway, ko00941 (Flavonoid biosynthesis), was enriched in all three combinations of CbW-vs-CbY, CbY-vs-CbO and CbO-vs-CbR in both histologies (Figure 7D). To determine how peppers are affected by related genes and metabolites during the color change process, we analyzed the expression of related genes and metabolites in Flavonoid biosynthesis. In the CbW-vs-CbY combination, 17 genes were up-regulated (*CQW23_29380*, *CQW23_09482* and *CQW23_19845*, etc.), 4 genes were down-regulated (*CQW23_25223* and *novel.807*, etc.), 7 metabolites were up-regulated (Butein, Phlorizin and Narigin, etc.) and 7 metabolites were down-regulated (Hesperetin, Luteolin and Tricetin, etc.). In CbY-vs-CbO, 2 genes were up-regulated (*novel.6029* and *CQW23_29123*), 11 genes were down-regulated (*CQW23_12748*, *CQW23_16084* and *CQW23_12748* etc.) and all metabolites showed up-regulated expression (12). In CbO-vs-CbR, all genes were expressed up-regulated (9), 16 metabolites were expressed down-regulated and only 1 metabolite (Dihydromyricetin) was up-regulated.

From the analysis of the above metabolome and transcriptome, we conclude that flavonoid metabolism and carotenoid metabolism play a crucial role in the trans-coloration of peppers. Combining the metabolomic and transcriptomic analysis of the KEGG enrichment pathway, we illustrate in more detail the expression of genes and metabolites associated with flavonoid metabolism and carotenoid metabolism throughout pepper development (Figure 8A,B).

Our data showed that genes *CQW23_16085*, *CQW23_16084*, *CQW23_19845*, *CQW23_24900*, *CQW23_29123* and *CQW23_29380* were highly expressed in the white fruit period and were expressed at low or no levels in the other three periods. The genes *CQW23_12748*, *CQW23_32878* and *CQW23_04979* were expressed at low or no expression in the white fruit period and at the highest expression in the orange fruit period. Naringnin chalcone and Pinobanksin were up-regulated in both CbW-vs-CbY and CbY-vs-CbO and down-regulated in CbO-vs-CbR; Luteolin and Tricetin were down-regulated in both CbW-vs-CbY and CbO-vs-CbR and up-regulated in CbY-vs-CbO (Figure 8A).

The genes *CQW23_09483*, *CQW23_11317*, *CQW23_19986*, *CQW23_09027*, *CQW23_30321* and *CQW23_17736* were mainly expressed during the CbO period, and to a lesser extent in CbY and CbR, and almost not in CbW; the genes *CQW23_05387*, *CQW23_06623*, *CQW23_14283* and *CQW23_28374* were mainly expressed during the CbW period (Figure 8B).

### 2.10. qRT-PCR Validation of Gene Expression Profiles

To validate our transcriptome sequencing results, we randomly selected 15 DEGs involved in carotenoids and flavonoids for qRT-PCR validation. Includes β-carotene hydroxylase (*CQW23_07725*), lycopene β-cyclase (*CQW23_09027*), capsaicin synthase (*CQW23_30321*), caffeoyl-CoA O-methyltransferase (*CQW23_32878*), carotenoid cleavage dioxygenase (*CQW23_00879*), cinnamic acid 4-hydroxylase (*CQW23_16084*), lycopene ε-cyclase (*CQW23_06623*), etc. The results showed that the DEGs in the qRT-PCR data were consistent with the sequencing results during the development of HNUCB0081 pepper fruit (Figure 9).

## 3. Discussion

The color of the fruit is considered to be the most important external feature for fruit selection and post-harvest life [17]. Pepper has gained its commercial value in the horticultural industry due to the unique pungent flavor of its fruits [18]. In plants, carotenoids and flavonoids usually give the fruit its different colors [19,20]. It has been suggested that mutations in genes in the carotenoid pathway may give tomato fruits an orange or yellow color [21]. Carotenoids are synthesized in plants through different metabolic processes with different biological functions [22,23]. Carotenoids have diverse functions in terms of antioxidant, light trapping, photoprotection, and prevention of degenerative diseases such as atherosclerosis, cancer and aging [24,25]. In 2019, Harriet M. Berry et al. have also demonstrated that the color intensity of pepper fruit is associated with the transcript level of the PSY-1 gene [26]. Flavonoids are secondary metabolites widely found in plants and are widely present in foods that people consume regularly, such as fruits, vegetables, and cereals [27,28,29]. It has been demonstrated that the color transformation of plant fruits such as tomatoes and citrus are associated with flavonoids [20,30,31,32]. In the present study, we used pepper HNUCB0081 (*C. baccutum*) fruits with four different trans-color stages of white, yellow, orange and red as experimental material for comparative analysis using targeted (carotenoids) and broad-target (flavonoids) metabolomes and transcriptomes to determine the material composition of pericarp color.

The results of the metabolomic analysis showed that 307 flavonoid metabolites and 68 carotenoid metabolites were identified in the pericarp of HNUCB0081. Among the 307 flavonoid metabolites, 10 secondary classification substances were included, of which the numbers of flavonols, flavones, and dihydroflavones were 118, 89, and 40, respectively. Our data indicate that the types of metabolites that are significantly enriched at different developmental periods in pepper fruit differ, with flavonol and flavones being the major metabolites in pepper fruit discoloration, and that Luteolin-7-O-(6″-malonyl) glucoside, 3,4′-Dihydroxyflavone and 5,6,7,4′-Tetramethoxyflavone are important flavonoid metabolites in the color change of pepper. Quercetin-3-O-arabinoside, Quercetin-3-O-rhamnoside (Quercitrin) and Isorhamnetin-3-O-sophoroside are important flavonol metabolites in the color change process of pepper. Flavonoids usually found in epidermal tissue, are the main molecules involved in plant pigmentation and accumulate through various modes during fruit development [33,34]. The metabolites of flavonoids and carotenoids mainly were up-regulated during the yellow-orange fruit period, while most of the metabolites of flavonoids and carotenoids were down-regulated during the orange-red fruit period, suggesting that the types and contents of metabolites differed during the different developmental stages of pepper fruit. We hypothesize that the production and transformation of metabolites occur mainly in the yellow-orange fruit period.

By RNA-seq analysis, in the flavonoid biosynthetic pathway, genes such as *CHS* and *C4H* were identified. CHS are key enzymes in the flavonoid biosynthetic pathway [35]. CHS was identified in 1972 as the first enzyme involved in the flavonoid biosynthetic pathway, and to date, *CHS* has been identified in many angiosperms [36,37]. In the present study, during the white fruit period, Cinnamoyl-CoA was catalyzed by C4H (*CQW23_16085* and *CQW23_16084*) to generate *p*-Coumaroyl-CoA, which is consistent with the findings of Lee et al. [38]. *p*-Coumaroyl-CoA was catalyzed by CHS (*CQW23_29123* and *CQW23_ 29380*) to produce Naringenin chalcone, which is consistent with the conclusion reached by Yin et al. [39]. *p*-Coumaroyl-CoA was subsequently catalyzed by CYP450 (*CQW23_19845*, *CQW23_24900*) to produce Caffeoyl-CoA, and Caffeoyl-CoA was catalyzed by CHS (*CQW23_ 29380*) to produce 2′,3,4,4′,6′-Peptahydroxychalcone 4′-O-glucoside. During the yellow and orange fruit periods, Caffeoyl-CoA was catalyzed by CCoAOMT (*CQW23_32878*, *CQW23_10085* and *CQW23_09726*) to produce Homoeriodictyol, which was subsequently converted to Eriodictyol, and finally to Tricetin. Key DEGs such as *CQW23_09483* (*PSY*), *CQW23_11317* (*PDS*) and *CQW23_19986* (*ZDS*), *CQW23_09027* (*LYC*), *CQW23_06623* (*LCYE*), *CQW23_05387* (*ZEP*), *CQW23_30321* (*CCS*) and *CQW23_17736* (*NCED*) were identified in the carotenoid synthesis pathway. During the transition from the GGPP molecule to β-carotene, the *CQW23_09483* (*PSY*), *CQW23_11317* (*PDS*) and *CQW23_19986* (*ZDS*) and *CQW23_09027* (*LYC*) genes were all largely significantly expressed in the yellow and orange fruit periods. It has been shown that *LYC* is capable of catalyzing the synthesis of orange alpha-carotene and beta-carotene [40]. In the β-carotene to Xanthoxin process, the *ZEP* (*zeaxanthin epoxidase*; *CQW23_05387*), *CCS* (*capsanthin/capsorubin synthase*; *CQW23_30321*), and *NCED* (*9-cis-epoxycarotenoid dioxygenase*; *CQW23_17736*) genes were significantly expressed in the orange and red fruit periods. During the production of Zeinoxanthin from Lycopene, CY*P450* (*cytochrome P450*; *CQW23_28374*) were significantly expressed during the white fruit period. Transcription factors also play an important role in plant growth and development as they regulate gene expression at the transcriptional level by binding to promoter regions [41,42]. In the present study, we identified Pinobanksin, Naringenin Chalcone and Naringenin as key metabolites in the flavonoid biosynthetic pathway, and Phytoene, Lycopene, β-carotene and ε-carotene as key carotenoid biosynthetic pathway metabolites. *CHS*, *C4H* and *CYP450* are key genes in the flavonoid biosynthesis pathway [43,44]. *PSY*, *PDS*, *ZDS*, *LYC*, *ZEP*, *NCED* and *CCS* are key genes in the carotenoid biosynthesis pathway [26,45]. In previous studies we learned that the color transformation of pepper fruit is closely related to the flavonoid and carotenoid content within the pepper fruit [4,46]. Therefore, we suggest that metabolites such as Pinobanksin, Naringenin Chalcone and Naringenin, Phytoene, Lycopene, β-carotene and ε-carotene are closely related to the mechanism associated with color change in pepper fruits.

Several transcription factors also regulate the biosynthesis of flavonoids and carotenoids. Studies have shown that the biosynthesis of flavonoids is regulated by the MBW complex, including *myeloblastosis* (*MYB*), *basic helix-loop-helix* (*bHLH*) and *WD-repeat protein* (*WD40*) [47]. The differentially expressed genes in the 3 comparative combinations of CbW-vs-CbY, CbY-vs-CbO, and CbO-vs-CbR in this study were annotated to a total of 4695 transcription factors belonging to 457 TF families, which mainly included Pkinase (345), NBARC (191), p450 (182) and UDPGT (78). Among the 17 major TF families with high numbers, the numbers of up- and down-regulated genes in the three comparative combinations were 1041 and 574, respectively, and there was a decreasing trend in the number of genes in the 3 combinations. The number of genes in the Pkinase family was the highest among the three comparative combinations, and it was shown that Pkinase could modify the plant response to osmotic and salt stress [48]. In addition, the number of transcription factors from the p450, NBARC, Pkinase_Tyr and Myb_DNA binding families is relatively high, and genes from these families may also play important roles in the color change of pepper fruits. Moreover, transcription factors such as *NBARC* (*CQW23_18627*) was identified in the first 20 genes in the pink module of the WGCNA analysis. It has been shown that p450 belongs to a family of heme-binding proteins that catalyze a variety of functional monooxygenase reactions involved in oxidative metabolism, and it has been shown that p450 is associated with hydrogen peroxide homeostasis [49,50], in the present study, genes of the *p450* family (*CQW23_16085*, *CQW23_16084*, *CQW23_24900*, *CQW23_19845*, *CQW23_28374*) play a key role in flavonoid and carotenoid biosynthesis.

## 4. Materials and Methods

### 4.1. Experimental Materials and Sampling

In this study, pepper HNUCB0081 (*C*. *baccatum*) grown at the Agricultural Science Experimental Base of Hainan University, China (110.328729 E, 20.056729 N). The pepper fruit undergoes 4 periods from germination to maturity: white fruit period (CbW, about 12 days after flowering), yellow fruit period (CbY, about 20 days after flowering), orange fruit period (CbO, about 25 days after flowering) and red fruit period (CbR, about 29 days after flowering) (Figure 1). Healthy pericarp samples from all four developmental periods of pepper were collected on 20 April 2021. Three biological replicates of each period sample were placed in 2 mL centrifuge tubes, rapidly frozen in liquid nitrogen, and stored at −80 °C for subsequent analysis.

### 4.2. Physiological Index Measurement

The total carotenoid, flavonoid and chlorophyll contents in the peel of pepper at four periods were determined by spectrophotometry using a plant carotenoid content assay kit, a plant chlorophyll content assay kit and a plant flavonoid content assay kit (Solarbio, Beijing, China, BC4330).

### 4.3. Metabolomic Analysis

Sample preparation, metabolite extraction and analysis, metabolite identification and quantification were carried out at Wuhan Metaville Biotechnology Co., Ltd. (Wuhan, China, www.metware.cn (accessed on 1 May 2022) according to their standard procedures as previously described by Zhou et al. 2022. A total of 12 pepper peel samples were freeze-dried by vacuum freeze-dryer (Scientz-100F). The samples were placed in a refrigerator at 4 °C overnight. After centrifugation at 12,000 rpm for 10 min, the extracts were filtered for UPLC-MS/MS analysis. Metabolites were analyzed qualitatively and quantitatively (MRM) using the MetWare Biotechnology Ltd. proprietary database (MWDB), mass spectrometry data were analyzed using the software Analyst 1.6.3, and metabolite content data were processed using UV (unit variance scaling) and thermographed by the R package ComplexHeatmap. The OPLS-DA model was developed using multiple supervised methods. The importance projections of the variables obtained for the multivariate analysis of the OPLS-DA model, as well as metabolites with Fold Change ≥ 2 and Fold Change ≤ 0.5 were considered as differential metabolites.

### 4.4. RNA-Seq Analysis

The transcriptome sequencings were commissioned from Novogene Technology Co. (Beijing, China). Total RNA was extracted from frozen pepper tissue using standard extraction methods, the RNA samples were rigorously quality controlled using the Agilent 2100 bioanalyzer, libraries were constructed using the NEBNext Ultra directional RNA library prep kit for illumina, and the mRNA libraries from each sample were subse-quently sequenced using Illumina. The low-quality reads, N-containing, and spliced reads were removed from the sequenced data. The Q20, Q30 and GC contents of the clean data were subsequently calculated. HISAT2 was used to map the paired-end clean reads to the reference genome for comparison. Novel gene prediction was performed using StringTie (1.3.3b) [51]. Differential expression analysis was performed using DESeq2 software (1.20.0) [52] to compare between combinations (genes with a *p*-value < 0.05 were assigned as differentially expressed) and GO and KEGG enrichment analysis was implemented.

### 4.5. WGCNA Analysis

We performed weighted gene co-expression network analysis using the R package (WGCNA) and selected a soft threshold for co-expression network clustering based on the FPKM values of all genes in the sample, choosing R^2^ > 0.8 as the criterion. All FPKM values were converted into a topological overlap matrix (TOM) and each gene was analyzed by hierarchical clustering [53]. The dynamic tree cutting method was used to divide different genes into different co-expression modules, merge the clustered similar co-expression modules, and calculate the correlation between different modules and the degree of association of genes within the modules. Key modules associated with pepper fruit color were found and the co-expression network was visualized for subsequent analysis using Cytoscape_V.3.7.1 software (MAC, Fresno, CA, USA).

### 4.6. RT-qPCR Analysis

We selected 15 genes from the flavonoid and carotenoid synthesis pathways for RT-qPCR validation. RNA was isolated from peel samples of pepper HNUCB0081 at four different developmental stages and used for transcription into cDNA. Actin was used as an internal reference gene and we used SYBR^®^ Select Master Mix (2X), Thermo NANO DROP 8000, BIO-RAD GelDocTMXR+ and Applied Biosystems StepOnePlusTM Real-Time System for the procedure of: 40 cycles of 95 °C for 30 s, 95 °C for 10 s and 60 °C for 30 s. Three biological replicates were made for each sample, and each plate was repeated 3 times in independent runs. The expression levels of the different genes were analyzed using the ΔΔCT method.

## 5. Conclusions

In the present study, we provide new insights into the mechanisms involved in the interaction of genes and metabolites involved in the biosynthesis of flavonoids and carotenoids in pepper fruit leading to color change in pepper fruit. We identified Pinobanksin, Naringenin Chalcone and Naringenin as key metabolites in the flavonoid biosynthetic pathway, catalyzed by the key genes *CQW23_29123*, *CQW23_29380* and *CQW23_12748*. In the carotenoid biosynthetic pathway, *CQW23_09483*, *CQW23_11317*, *CQW23_09027* and *CQW23_30321* catalyze the synthesis of key metabolites such as Phytoene, Lycopene, β-carotene and ε-carotene in the carotenoid biosynthesis pathway. Fruit color change in peppers, especially in peppers with multiple color change periods, is a complex multi-omics and multi-material interaction process, and further investigations are needed to fully understand the molecular mechanisms and principals involved in each of these steps.

## Figures and Tables

**Figure 1 ijms-23-12524-f001:**
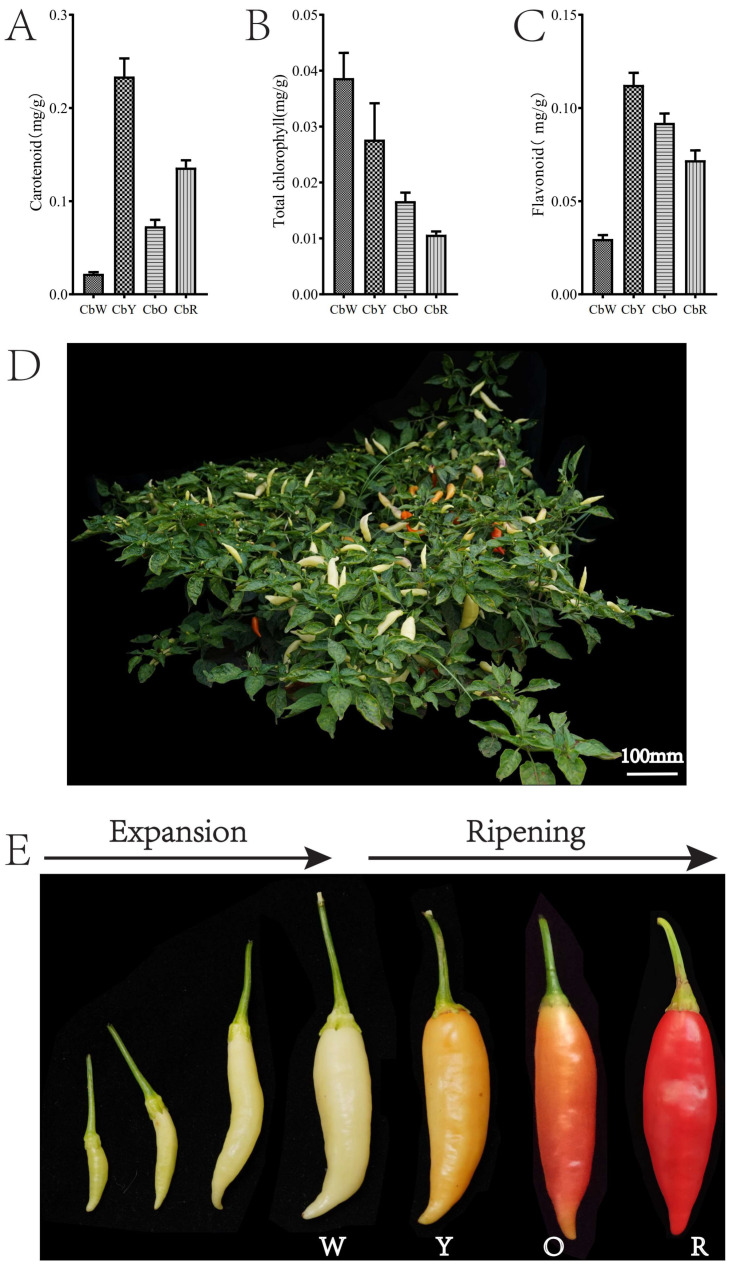
Plant, fruit and physiological index contents of pepper HNUCB0081. (**A**–**C**) Total carotenoids, chlorophylls and flavonoids in fruits of pepper at four periods; (**D**) plants; (**E**) fruits at different developmental periods (W: CbW, Y: CbY, O: CbO, R: CbR).

**Figure 2 ijms-23-12524-f002:**
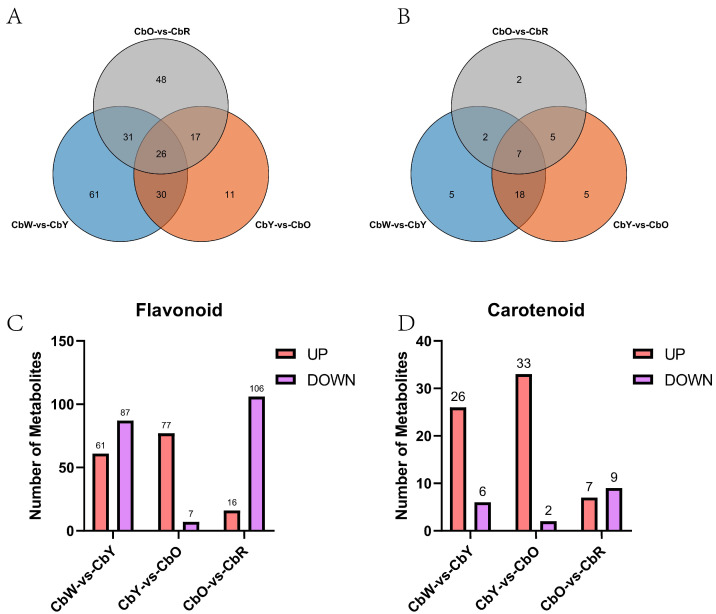
Metabolite analysis in CbW-vs-CbY, CbY-vs-CbO and CbO-vs-CbR. (**A**) Total number of flavonoid metabolites; (**B**) total number of carotenoid metabolites; (**C**,**D**) number of up- and down-regulated metabolites of flavonoids and carotenoids.

**Figure 3 ijms-23-12524-f003:**
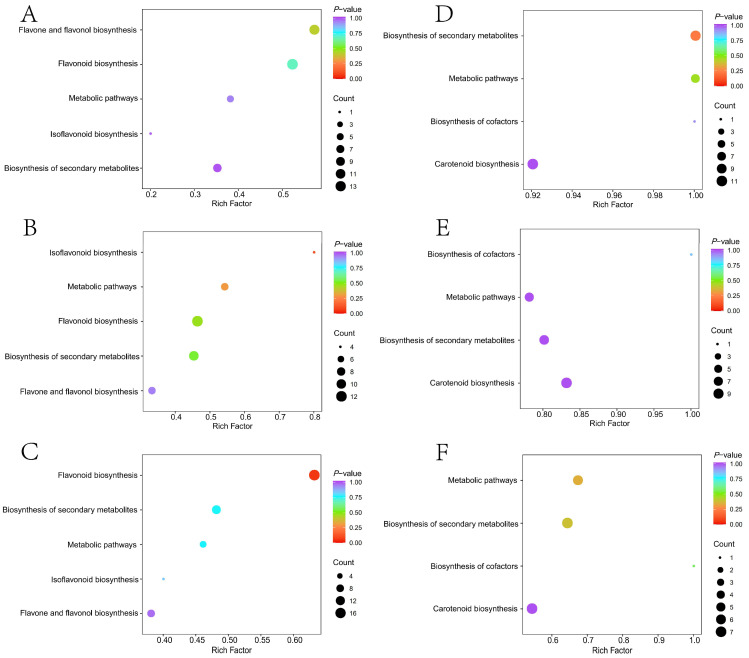
Metabolite analysis. (**A**–**C**) Differential metabolite enrichment pathways of flavonoids in CbW-vs-CbY, CbY-vs-CbO, and CbO-vs-CbR.; (**D**–**F**) Differential metabolite enrichment pathways for carotenoids in CbW-vs-CbY, CbY-vs-CbO, and CbO-vs-CbR. Note: The horizontal coordinate indicates the corresponding rich factor for each pathway, the vertical coordinate is the pathway name, and the color of the dots is *p*-Value, with more red indicating more significant enrichment. The size of the dots represents the number of differential metabolites enriched.

**Figure 4 ijms-23-12524-f004:**
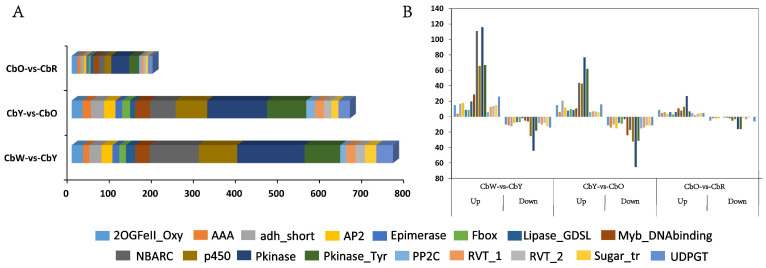
Transcription factor analysis. (**A**) Distribution of transcription factor families in the three comparative combinations of CbW-vs-CbY, CbY-vs-CbO and CbO-vs-CbR. (**B**) Number of up- and down-regulated transcription factor family members in three comparative combinations of CbW-vs-CbY, CbY-vs-CbO and CbO-vs-CbR.

**Figure 5 ijms-23-12524-f005:**
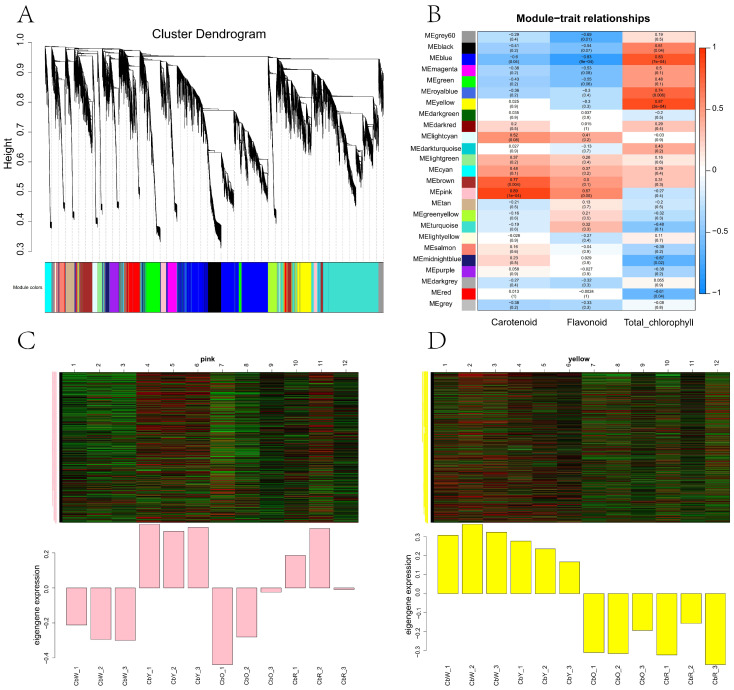
(**A**) Module hierarchy clustering tree diagram. (**B**) Heat map of correlations between modules. (**C**,**D**) Heat map of correlations between sample traits and modules, (**C**) is the pink module and (**D**) is the yellow module. The horizontal coordinate is the sample, the vertical coordinate is the module, the number in each grid represents the correlation between the module and the sample, the closer the value is to 1, the stronger the positive correlation between the module and the sample; the closer it is to −1, the stronger the negative correlation between the module and the sample. The number in parentheses represents the significance Pvalue, the smaller this value is, the stronger the significance is. def: the amount of gene expression within the module.

**Figure 6 ijms-23-12524-f006:**
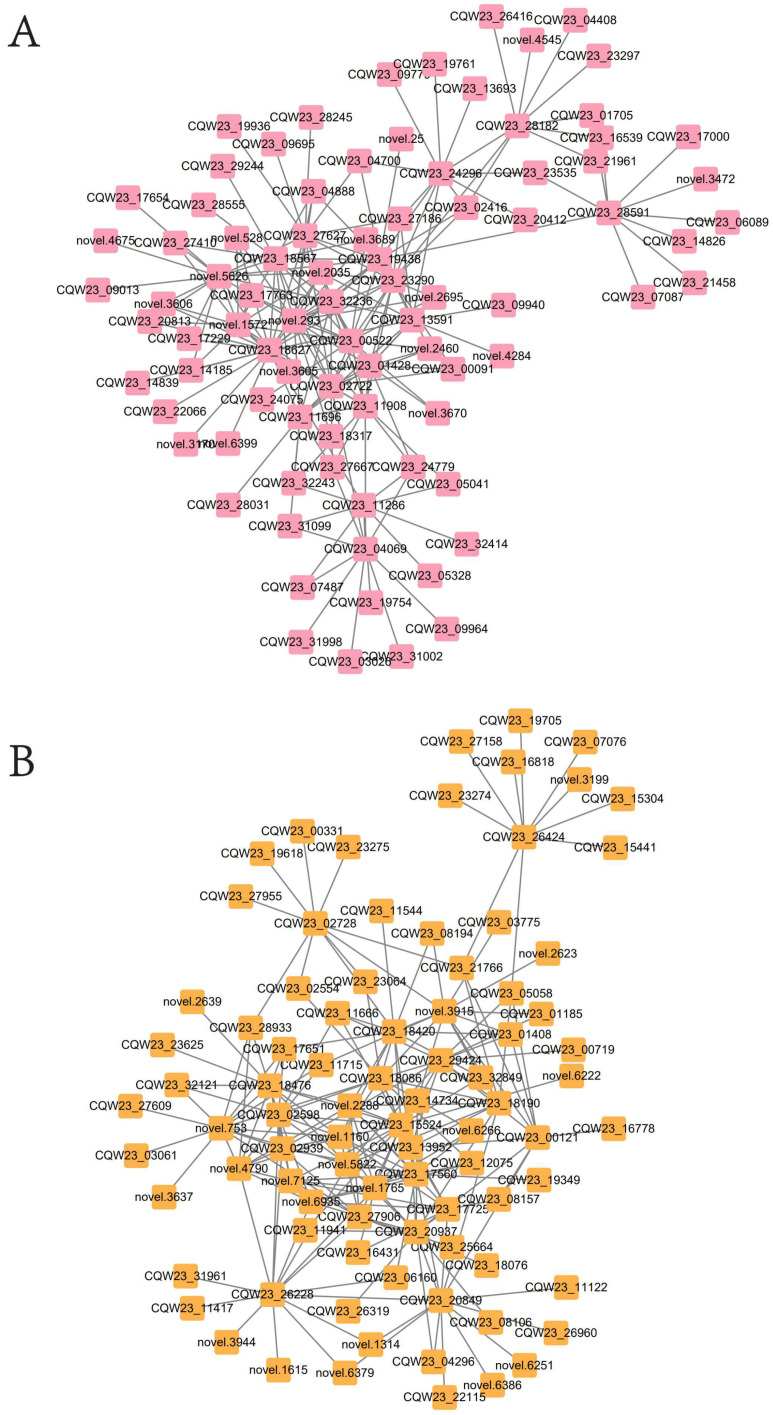
(**A**) Pink module genes correlation network; (**B**) yellow module genes correlation network.

**Figure 7 ijms-23-12524-f007:**
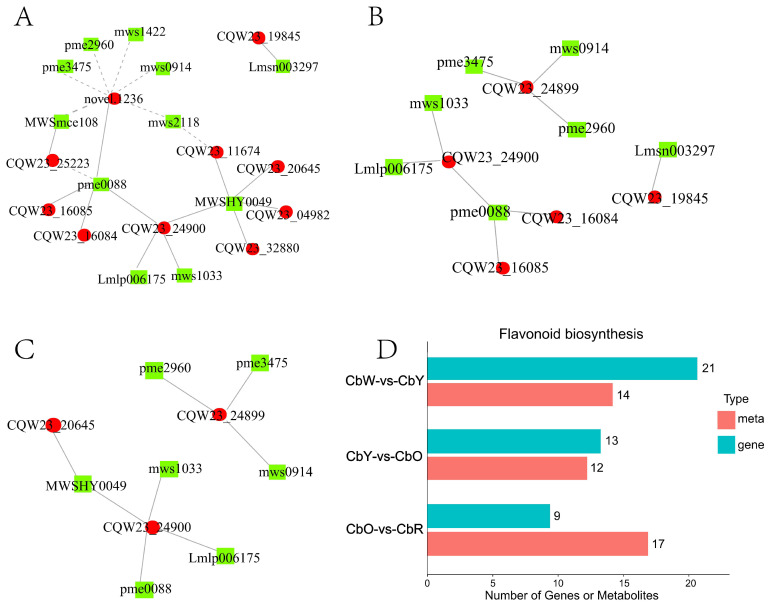
Correlation analysis of transcriptome and metabolome. (**A**–**C**) Interactions between genes and metabolites in CbW-vs-CbY, CbY-vs-CbO and CbO-vs-CbR; (**D**) pathways co-enriched to genes and metabolites in CbW-vs-CbY, CbY-vs-CbO and CbO-vs-CbR.

**Figure 8 ijms-23-12524-f008:**
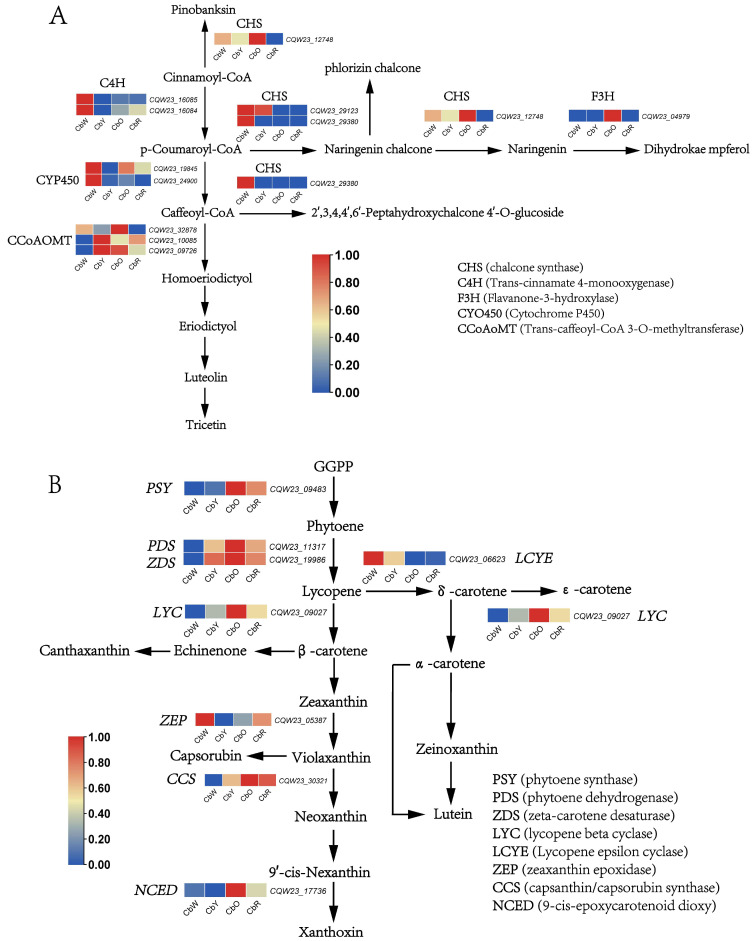
Flavonoid and carotenoid biosynthesis pathway diagrams. (**A**) Diagram of flavonoid biosynthesis pathway; (**B**) Diagram of carotenoid biosynthesis pathway. (This diagram was drawn based on references to relevant literature and data in this study).

**Figure 9 ijms-23-12524-f009:**
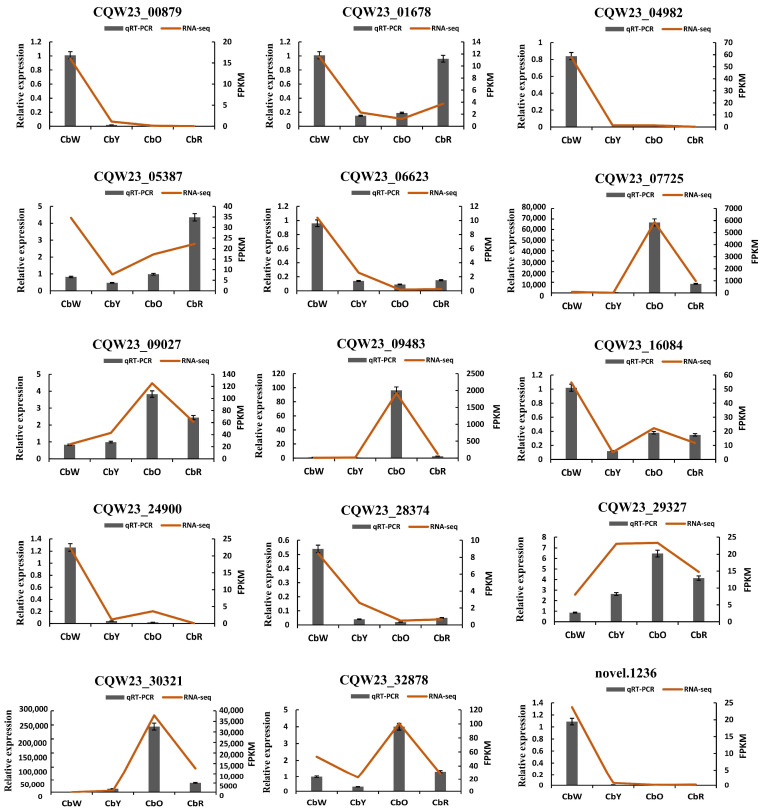
The expression profiles of significantly differentially expressed genes were measured by RNA-seq and qRT PCR. The histogram data indicate qRT-PCR data (left *Y*-axis), dashed line indicates RNA-seq data (right *Y*-axis).

## Data Availability

Supporting reported data can be found at https://dataview.ncbi.nlm.nih.gov/object/PRJNA791157 (accessed on 29 August 2022).

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
