# Peer review of "Transcriptome and Metabolome Analysis of Color Changes during Fruit Development of Pepper (Capsicum baccatum)"

_ijms, 2022, doi:10.3390/ijms232012524_

Round 1

Reviewer 1 Report

Aesthetics is always one to consider, since the color of fruit may also be directly linked to how rich in certain nutrients they are. Research work under consideration tried to identify genetic elements involved in Pepper fruit color change through multiomic techniques. Overall research work is convincing, and manuscript is well crafted. However, certain analytical questions remain unanswered that are listed below.

1.     Several Flavonoid and Carotenoid metabolites were identified such as Pinobanksin, Naringenin Chalcone, Phytoene, Lycopene, β -carotene and ε -carotene as key metabolites. However, authors failed to mention the key metabolites responsible for color change in pepper fruit.

2.     Through WGCNA (weighted gene co-expression network analysis) analysis, several modules are positively correlated with flavonoids and carotenoids, on what basis most related module was selected for further analysis.

3.     Transcription factors play vital role in the regulation of color change, this research specified the top 20 genes of connectivity as hub genes. Were there any transcription factor identified as the hub genes connecting to biosynthesis and transporter gene of flavonoids and carotenoids.

4.     On what indices fruits were selected as of different stage?

5.     Several factors such as environment and light, affect fruit coloring and maturity, what were the growth conditions and were fruits of different stages were collected from same plants?

6.     How did the author select specific candidate genes among tens of candidates? In the discussion, authors should describe a relevance between candidate genes and phenotypes. I don't see any reasonable logic to select the genes.

7.     Generally, the manuscript is well written but there is a need to improve discussion with reasoning and supportive literature.  

Other comments

1.     The authors has not provided the line number in the manuscript, so it is difficult to point out particular improvement in the manuscript.

2.     There are some typographical error in the manuscript which needs to be rectified.

3.     Fig. 8 and 9 caption needs to be explained in detail.

Reviewer 2 Report

Crop's color change is very important agronomic trait for increasing product value. Your research is worthy for this point, although carotenoid and flavonoid biosynthesis for color change is not considered a new study. So, I expect that your study would be a good reference in the future study and I'd like to accept your paper with minor revision as follows:

1. Improve English writing. Please represent well full-name of each abbreviation, even in figure legends in order.

2. State more the importance of your research, carotenoid and flavonoid biosynthesis involved in color change.  

Your paper is generally well-written, so I don't think there is more to point out. Thank you for your good study.

Round 2

Reviewer 1 Report

I think the authors made substantial modifications to the manuscript according to the comments. Best wishes.